# Digital Optical Ballistocardiographic System for Activity, Heart Rate, and Breath Rate Determination during Sleep

**DOI:** 10.3390/s22114112

**Published:** 2022-05-28

**Authors:** Nuria López-Ruiz, Pablo Escobedo, Isidoro Ruiz-García, Miguel A. Carvajal, Alberto J. Palma, Antonio Martínez-Olmos

**Affiliations:** ECsens, CITIC-UGR, Department of Electronics and Computer Technology, University of Granada, 18071 Granada, Spain; nurilr@ugr.es (N.L.-R.); pabloescobedo@ugr.es (P.E.); isirg@ugr.es (I.R.-G.); carvajal@ugr.es (M.A.C.); ajpalma@ugr.es (A.J.P.)

**Keywords:** ballistocardiogram, digital detector, instrumentation, android application

## Abstract

In this work, we present a ballistocardiographic (BCG) system for the determination of heart and breath rates and activity of a user lying in bed. Our primary goal was to simplify the analog and digital processing usually required in these kinds of systems while retaining high performance. A novel sensing approach is proposed consisting of a white LED facing a digital light detector. This detector provides precise measurements of the variations of the light intensity of the incident light due to the vibrations of the bed produced by the subject’s breathing, heartbeat, or activity. Four small springs, acting as a bandpass filter, connect the boards where the LED and the detector are mounted. Owing to the mechanical bandpass filtering caused by the compressed springs, the proposed system generates a BCG signal that reflects the main frequencies of the heartbeat, breathing, and movement of the lying subject. Without requiring any analog signal processing, this device continuously transmits the measurements to a microcontroller through a two-wire communication protocol, where they are processed to provide an estimation of the parameters of interest in configurable time intervals. The final information of interest is wirelessly sent to the user’s smartphone by means of a Bluetooth connection. For evaluation purposes, the proposed system has been compared with typical BCG systems showing excellent performance for different subject positions. Moreover, applied postprocessing methods have shown good behavior for information separation from a single-channel signal. Therefore, the determination of the heart rate, breathing rate, and activity of the patient is achieved through a highly simplified signal processing without any need for analog signal conditioning.

## 1. Introduction

The use of ballistocardiography (BCG) as a technique for evaluating the cardiac function of a patient was first reported in the middle of the twentieth century [1], although the movements imparted to the human body by the impacts of the blood were first recorded several decades earlier [2]. Ballistocardiographs were originally based on the longitudinal displacement experienced by a rigid table where the patient was lying due to the impulses of their heart. The table, constructed of a thin three-ply panel mounted on a spruce frame and braced by a steel truss, with a weight of 50 pounds, was supported by a strong steel spring, and the longitudinal movement was optically registered by making use of a lamp and a camera [1]. It was found that the size and shape of the ballistic curves are determined by the blood velocity in the great vessels, and therefore, they are useful tools in identifying patients with abnormal circulation or heart diseases [3].

BCG was the objective of extensive research until the mid-1970s. Since then, interest in this technique has faded away due to a variety of reasons, such as the lack of standard measurement techniques and the appearance of ultrasound and echocardiography techniques for cardiac and hemodynamic diagnostics [4]. Nevertheless, in the last decade, the research and use of BCG have gained renewed attention due to the development of new sensors and computational techniques, which have led to more accurate results. At the present time, contact and non-contact BCG sensors can be used with no need for technical or medical staff. In addition, BCG is a non-invasive and comfortable procedure for the patient that helps reduce the stress caused by other medical equipment, such as an electrocardiograph [5].

Numerous recent studies have been reported where a variety of sensors has been applied to the recording of BCG curves, including piezoelectric polyvinylidene fluoride (PVDF) [6,7], electromechanical film (EMFi) [8,9], pneumatic- and hydraulic-based sensors [10,11,12], strain-gauges-based sensors [13,14], and fiber-optic-based sensors [15,16], among others.

The application field of BCG has also been extended in these years. Initially, this technique was traditionally oriented to clinical applications [17], and in particular, to the diagnosis of heart diseases. Today, it is still one of its main objectives. Furthermore, since BCG signals can also accurately detect human respiratory status and predict respiratory diseases [18], the use of these systems is widely applied in several fields outside the medical scope. For instance, the use of BCG for the development of a health monitoring chair was reported [8,19]. In the automotive industry, this technique is being studied as a tool to measure the vital functions of occupants [20]. Sleep evaluation is one of the most promising application fields of BCG. It has been proven that the use of heart rate (HR), breathing rate (BR), and movement information of a sleeping subject can provide predictions of their sleep state [21]. These parameters can be obtained using BCG registers. Therefore, many BCG systems have been presented that are oriented to determining sleep parameters and the sleep stage of patients [22,23].

It is common to find that most BCG systems are composed of complex conditioning electronics with stages for the transduction, amplification, filtering, and leveling of the signals since the output of the sensors is usually a low-power analog signal.

In this work, we present a novel BCG system to be applied in beds with the aim of extracting HR, BR, and movement information of a lying patient that reduces the complexity of the usual acquisition electronics by making use of a commercial digital optical sensor. This device registers variations in the intensity of light reaching its sensing surface that are produced by vibrations of the bed mattress. The output of the detector is directly transmitted as digital words, and therefore, only digital processing is required to obtain estimations of the parameters of interest, thus avoiding the use of complex analog signal processing electronics. These results are presented to the user in a smart device by means of a custom-developed application. Therefore, the developed equipment is a non-invasive and easy-to-use system that consists of a simple sensor module and electronic board, together with a battery and an Android application.

## 2. Description of the System

A simplified block diagram of the proposed system is presented in Figure 1.

### 2.1. Electronics Design

The BCG prototype is mainly composed of two units: the sensing module and the microcontroller board. The sensor unit is the part that is sensitive to the vibrations of the mattress produced by the heartbeat, breathing, and movement of the patient during sleep. Although direct optical sensors are not often used in this technique, some attempts to register BCG curves by means of photodiodes and reflected light have been previously reported [24]. Here, we propose the use of a sensing module consisting of a white LED facing a digital photodetector, as depicted in Figure 2, where variations of direct light are directly registered.

The LED and the photodetector are placed in two boards of dimensions 5 × 5 cm^2^ coupled to each other by means of four small springs acting as buffers, as depicted in Figure 2. This module has a weight of 15 g. The springs are stainless steel, with a spring rate of 2 N/mm, and are 4.63 mm in diameter. Under stable biasing of the LED, the intensity of the detected white light only depends on the distance between the light source and the photodetector. This distance, which is fixed at 0.5 cm in the uncompressed mode, varies because of the compression of the springs when pressure is applied to the mattress.

The LED model GW PSLMS1.EC−GSGU−5C7E−1 (OSRAM Opto Semiconductors, Regensburg, Germany) is biased in the range of 0−10 mA using a programmable current source, as shown in the detail of Figure 2. The objective of biasing the LED with a selectable current is to implement an automatic optimization in the output of the digital photodetector for subjects with very different weights and constitutions. This avoids oscillations of the signal close to the limits of the allowed range, which might produce the saturation of the detector. The current flowing through the LED is determined by controlling the current across the resistor. The operational amplifier model LT1366 (Linear Technology Corporation, Milpitas, CA, USA) forces a voltage drop of VDAC in the resistor, with this value being the voltage generated in the digital to analog converter (DAC) of the microcontroller board. Considering that the bias voltage of the LED is very close to 2.62 V, and assuming a negligible voltage drop VDS in the NMOS transistor model 2N7000 (Semiconductor Components Industries LLC, Phoenix, AZ USA), a maximum value of VDAC of 0.68 V is allowed; therefore, using a resistance of 68 Ω, the maximum selectable current is 10 mA.

The digital detector used in the sensor module is the model S11059−02DT (Hamamatsu Photonics K.K., Hamamatsu, Japan), which is able to detect the intensity and color of the incident light, generating an output that consists of four 16-bit digital words corresponding to the red (R), green (G), blue (B), and infrared (IR) components of the incoming radiation. These words are serially transmitted through a two-wire connection (I2C protocol) to the microcontroller board. The photodiode for each color is automatically switched sequentially to perform the measurements. The integration time of the detector can be configured from microseconds to hundreds of seconds, thus allowing the detection of a wide range of light intensities. The use of this commercial sensor allows for the simplification of the scheme of the electronic design since the typical stages found in analog-sensor-based BCG systems, ‘transducer—amplifier—filter—range adaptor—analog-to-digital converter’, are performed within the device. It directly provides the digital samples related to the detected light, thus greatly reducing the need for processing electronics. Internally, the S11059 device contains a light-to-frequency converter. In addition, this module integrates not one but four photodiodes with different spectral sensitivities. This allows the selection of the band where the response of the sensor shows better behavior in terms of error and linearity, as explained in Section 3.1.

The microcontroller board is the commercial module model ESP32 (Espressif Systems Co., Ltd., Shanghai, China), which includes an Xtensa 32-bit LX6 microprocessor (Tensilica, Inc., Santa Clara, CA, USA). It was selected due to its powerful processing capabilities, as well as its multiple peripheral interfaces, such as programmable GPIOs, DAC, I2C, Bluetooth, and WiFi. In addition, the easy programming of this module is possible thanks to its compatibility with the Arduino Integrated Development Environment (IDE) (Arduino LLC, Somerville, MA, USA). This device receives the output of the digital color photodetector, traces the BCG curve from these results, and processes the data to generate estimations of the HR, BR, and activity or movement of the lying patient. It also provides a power supply to the sensing module and an analog voltage VDAC by means of the integrated DAC, which is the input value of the previously described programmable current source for the LED biasing. Finally, the ESP32 board transmits the data collected during the sleep of the subject to a smart device (smartphone or tablet) by means of a wireless Bluetooth connection.

### 2.2. Android Application

A user-friendly Android application was developed for the communication, data acquisition, and result displaying of the BCG platform. The application was programmed using the official Android IDE, Android Studio 4.2.2. It was designed and tested against API 28 (Android 9.0), although it also supports previous Android versions, including API 19 (Android 4.4), which is the lowest level compatible with the application.

The developed application implements the Bluetooth interface for communication with the ESP32 module, which contains a unique address assigned for the connection with the smart device (smartphone or tablet) via Bluetooth. When both devices are within range of each other and correctly paired, a new screen appears, consisting of an upper menu bar with three tabs, easily identifiable by icons corresponding to (i) heart rate, (ii) breath rate, and (iii) night activity (Figure 3). From this screen, the user can click on the ‘Receive Data’ button, which sends a command to the ESP32 module to start receiving all of the logged data for each measured parameter. The application continuously saves the information sent by the ESP32. Simultaneous to the data reception, the application generates text .csv files to save all data on the internal memory of the device. Once received, the data are plotted in every tab, as observed in Figure 3. The user can navigate through the plotted data with multitouch gestures such as pinch to zoom, as well as select any data point to see the value at a particular time. MPAndroidChart open-source charting library version 3.1.0 was used for data visualization within the application. Finally, the user can save screen captures of the graphs through the ‘Save to Gallery’ option in the upper menu bar of the application. From this menu, it is also possible to load previous .csv files stored on the phone and visualize the corresponding graphs from previous experiments carried out and saved in the smart device.

## 3. Results and Discussion

The presented BCG system was tested and validated using an individual mattress with a surface of 180 × 90 cm^2^ and 21 cm in height, along with a topper 3.5 cm thick. The sensing module was placed between the mattress and the topper at a middle thoracic position, as schematized in Figure 4.

The BCG system was configured to acquire data of light intensity with a sampling rate of 100 samples per second, the integration time of the digital detector being 1.4 ms per channel for system sensitivity optimization.

### 3.1. Static Mode

The error or uncertainty of the system was evaluated by collecting output data of the detector in static mode (i.e., with no subject lying on the bed) for 5 min. The sequences acquired for different bias currents of the LED were analyzed to determine the relative error as the standard deviation divided by the mean value of the sequence for each color [25]. Table 1 shows the obtained results. As it can be derived, lower errors are obtained for the B and IR channels.

The mean values of the sequences were also evaluated. Figure 5 presents the normalized mean intensities of each color component for different bias currents.

The IR component of the detected light shows a linear behavior when the bias current varies in the range of 0 to 10 mA. This result, together with the lower errors shown in Table 1, led us to use this component for the tracing of the BCG curves.

### 3.2. BCG Curves

As expressed in previous sections, the aim of this prototype is to measure oscillations of the mattress by registering variations in the light intensity detected by the digital photodetector. These intensity changes are caused by alterations in the distance between the white LED and the detector, which are produced by the mechanical oscillations that compress and relax the springs in the sensing module. In turn, these vibrations are generated by the forces of the heart beating, breathing, and movements of a lying patient. Therefore, the recording of the variation of the detected light intensity results in the trace of BCG curves.

Figure 6 shows examples of BCG curves obtained with the presented system corresponding to a healthy patient lying in supine (A) and prone (B) positions. Simple visual inspection of these graphics allows the identification of the peaks corresponding to the heart beating and the oscillation of the baseline due to the breathing of the subject. The curve shown in Figure 6B includes a period of several seconds where apnea occurred when the subject was asked to hold his breath. As expected, the movement of the subjects while they are lying on the bed causes vibrations in the mattress much stronger than those produced by heart beating and breathing, as can be seen in Figure 6C.

In Figure 7, a BCG signal obtained with the presented system is compared with the same curve obtained by means of a BCG system based on a strip consisting of an 800 mm long, 8 mm wide Piezo film (piezoelectric PVDF polymer, TE Connectivity, Ltd., Switzerland), also developed by our research group following the analog scheme proposed [26]. As is known, the strings compressed under the mass of the mattress and the patient act as mechanical isolators [27]. This produces an attenuation of the response of the system to vibration frequencies higher than the undamped natural frequency of the springs, which, in this case, is set close to 1 Hz, depending on the mass of the lying subject. As a result, the proposed system is mainly sensitive to the main frequencies of the mattress caused by the heartbeat and breathing of the subject, and the generated signal does not fit the classical BCG signal shape, as shown in Figure 7B [5]. This means that the developed prototype based on the digital optical detector is not able to provide information about the overall performance of the circulatory system. Nevertheless, from the curve of Figure 7B, the identification of the heart beating is simpler than in Figure 7A, not only by visual inspection but also by conducting a simpler signal processing than is usually included in BCG systems [21,28].

When the Fast Fourier Transform (FFT) is applied to both signals, the power spectrum at 1.2 Hz (which is the heart beating rate in this case) is five times higher for the signal obtained with the novel system than the power for the signal generated with the PVDF sensor. This implies that the proposed system is more sensitive to the main frequency of the heart beating and allows a better determination of this rate.

### 3.3. HR, BR, and Activity Determination

The output of the digital photodetector is continuously registered by the microcontroller board in order to generate the BCG curves. The acquired samples are stored and processed in configurable time intervals to generate an average estimation of the HR, BR, and movement in every interval. Given the memory restrictions of the microcontroller board, it is guaranteed that a minimum period of 8 h of test can be recorded with time intervals of 15 s.

To obtain the estimation of the mean BR within the interval, the samples are first filtered using a ninth-order low-pass digital infinite impulse response (IIR) filter with a pass-band frequency of 0.5 Hz and a stop-band frequency of 1.5 Hz. In this way, only low-frequency components of the signal corresponding to the waving of the baseline remain, as can be seen in the example of Figure 8 (blue line). As previously stated, such low variations of the signal are expected to be produced only by the breathing of the subject. After filtering, the FFT is applied to determine the frequency components of the sequence [29]. First, the element of the generated vector is discarded, since it corresponds to a DC component (signal offset). The higher element of the remaining data corresponds to the breathing frequency or rate of the subject.

For the estimation of the mean HR, the sequence of the BCG curve is double-filtered: first, a digital seventh-order IIR high-pass filter with a stop-band frequency of 0.1 Hz and a pass-band frequency of 0.5 Hz is used to eliminate low-frequency components caused by breathing; after that, another seventh-order IIR low-pass filter with a stop-band frequency of 1 Hz and a pass-band frequency of 5 Hz is applied to the data with the aim of reducing the frequency spectrum to the first harmonics (see red line in Figure 8). The main element of the FFT of these processed data corresponds to the mean BR in this time interval.

The presence of movements of the patient in the time interval being monitored has been evaluated by other authors through the variance measurement of the processed data [30]. The higher amplitude of the vibrations in the mattress produced by the movements of the subject produces a lower dispersion of the data. Therefore, the variance measure of sequences including these effects is expected to be lower than that of the clean sequences. In this work, we tried to use this single parameter to predict the presence of movements in the interval, but the hit rate reached was low. Moreover, this rate decays if the activity in the sequence is high, that is, if there are several movements of the subject in the interval analyzed, since the value of the variance measurements increases. Nevertheless, we found that the number of samples in the processed sequence with a value above the variance measure is much higher in a clean interval than that in a motion-corrupted one. This parameter allows us to predict the presence of motion in an interval with a hit rate very close to 100%, even if the activity in the sequence is high.

The processed data used to obtain this number of significative samples are generated from the original BCG curve after applying the double filter explained above for the prediction of the HR. The resulting data are squared to maintain a positive sign of the samples. The variance measure of this resulting sequence is calculated, and the number of samples with a value higher than this variance is obtained. We defined the density of significative samples as this calculated number of samples divided by the length of the sequence. This parameter is close to one in clean sequences, while it decreases to levels close to zero in sequences containing activity from the patient. In this work, we simply assumed that if this density is higher than 0.5, the sequence is clean; otherwise, the sequence is motion-corrupted.

In Figure 9, an example of the processed sequences for the calculation of the density of significative samples in an interval of 30 s is offered for a clean and corrupted sequence. As it can be observed, in the clean interval, there are many more significative samples over the variance value than in the second case, where a movement of the subject has occurred. The density of significative samples obtained in these examples is 0.79 and 0.09, respectively.

The use of this parameter allows us not only to predict if a movement has occurred but also the moment in time when it has taken place.

In order to illustrate the simplicity of the proposed system for the recording of BCG curves and the extraction of HR, BR, and activity, Table 2 shows a summarized comparison between the presented work and others recently published in the literature for the same purpose.

## 4. Conclusions

A novel scheme of a bed-based BCG monitoring system is herein presented, with simplicity being its main characteristic. The use of small springs for the transmission of mattress vibrations to an optical sensing module has proven to be a useful tool in the tracing of BCG curves. The detection of these vibrations by means of a digital photodetector has allowed for the reduction in the complexity of the sensing electronics while maintaining high system sensitivity. Moreover, the compressed springs act as mechanical isolators that attenuate the response of the system to frequencies higher than approximately 1 Hz. Therefore, the obtained curves do not show the classical BCG waves generated by heart activity, in which it is difficult to identify the single heartbeats. Instead, with the proposed system, a quasi-square-shaped signal is generated following the heartbeat, and it is possible to determine the parameters of interest, especially HR, with simplified hardware and software. Therefore, an approach for the estimations of the mean value of HR and BR in configurable time intervals has been proposed based on the generation of the FFT of the sequences in such intervals. Additionally, a novel technique for the determination of the presence of movement of the subject in the BCG curve is exposed. This technique is based on the calculation of the number of samples of the processed data whose value is higher than the variance measure, and it has been proven to predict the presence of movements and the moment in time when they occur with a high hit rate.

## Figures and Tables

**Figure 1 sensors-22-04112-f001:**
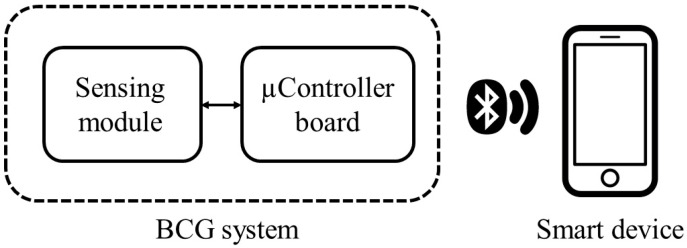
Block diagram of the system.

**Figure 2 sensors-22-04112-f002:**
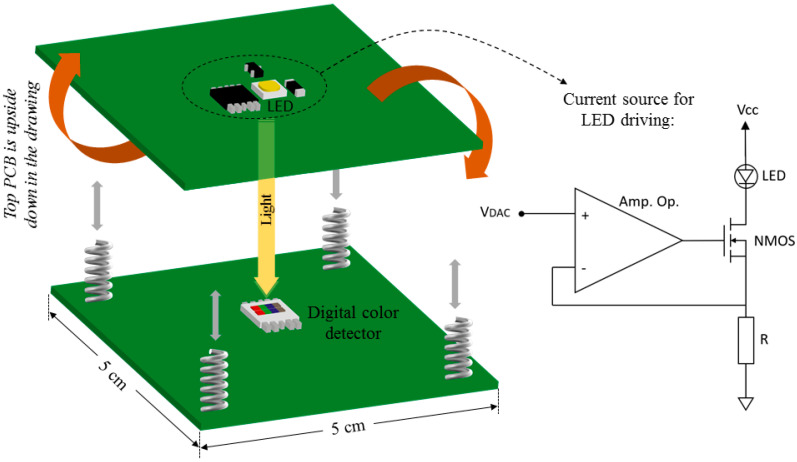
Sensing module and schematic of the current source.

**Figure 3 sensors-22-04112-f003:**
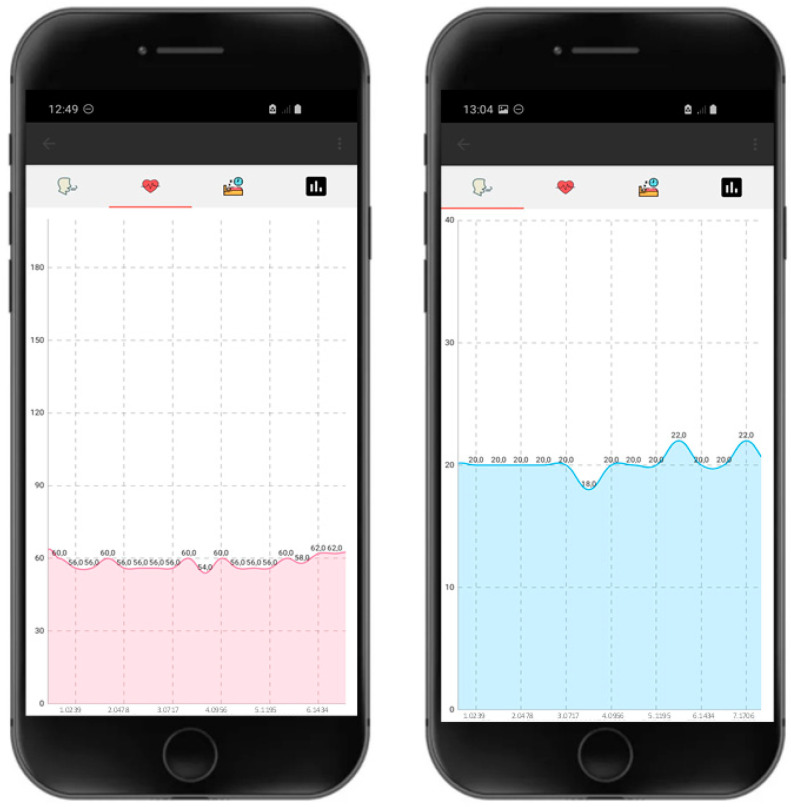
Screen captures of the custom-developed Android application showing examples of plotted data corresponding to heart rate (**left**) and breath rate (**right**).

**Figure 4 sensors-22-04112-f004:**
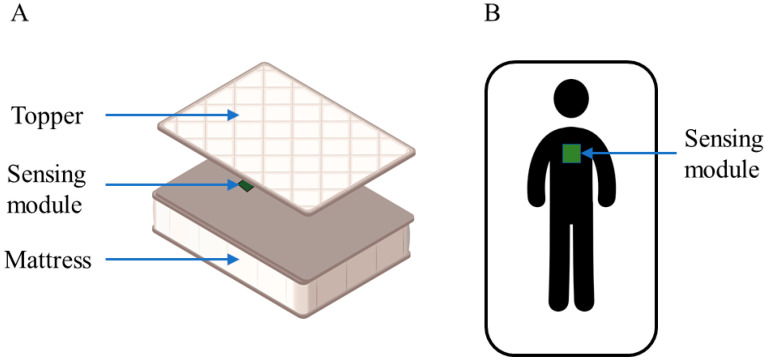
Location of the sensing module in the bed: perspective view (**A**) and top view (**B**).

**Figure 5 sensors-22-04112-f005:**
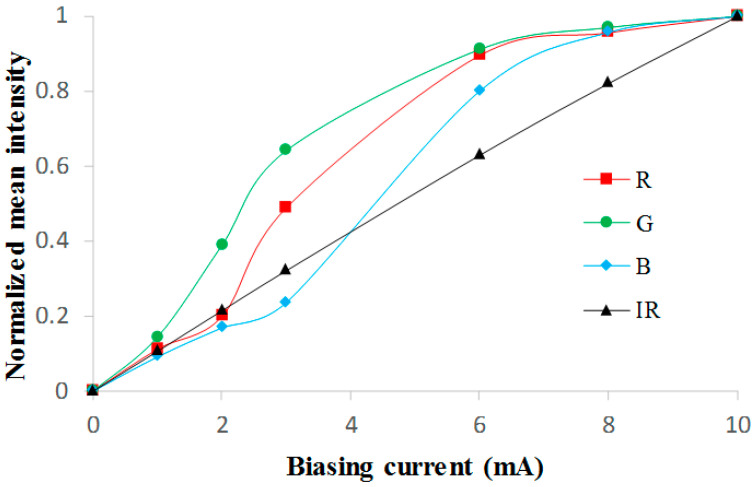
Normalized mean light intensities for bias currents in the range 0–10 mA.

**Figure 6 sensors-22-04112-f006:**
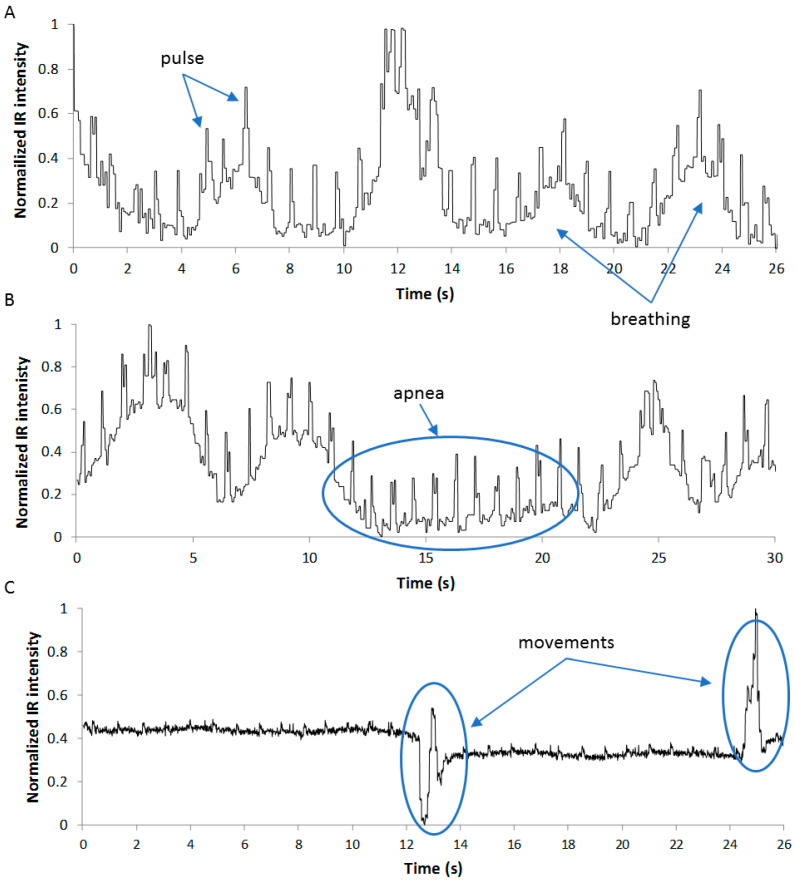
Example of obtained BCG curves. Subject in supine (**A**) and prone (**B**) positions; sequence corrupted by the activity of the patient (**C**).

**Figure 7 sensors-22-04112-f007:**
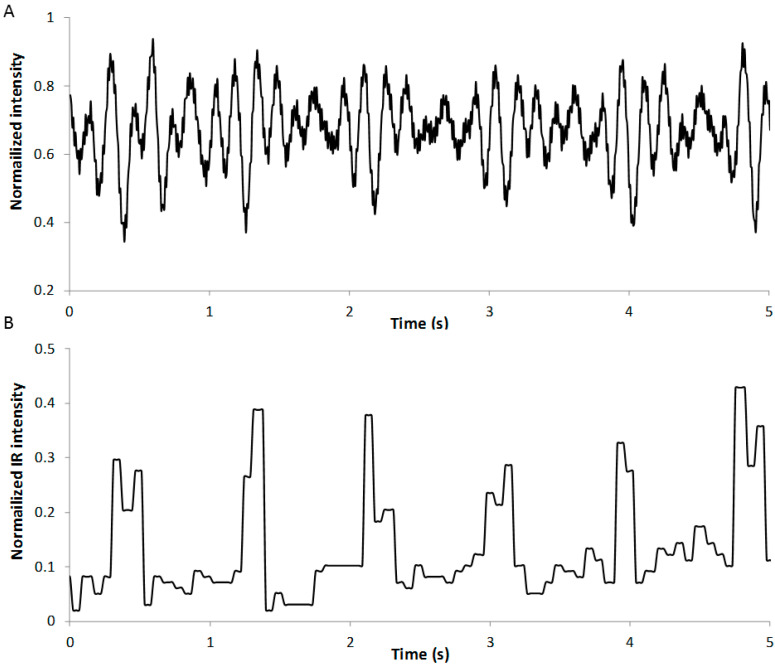
Comparison of BCG curves obtained with a PVDF sensor (**A**) and with the proposed optical sensor (**B**).

**Figure 8 sensors-22-04112-f008:**
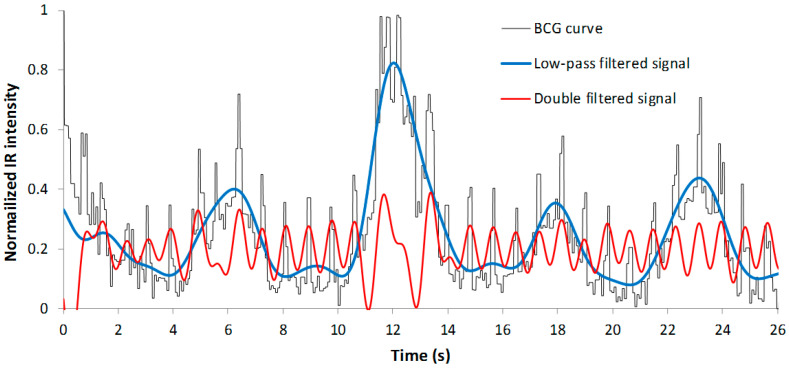
Signals generated after filtering the BCG curve for the prediction of the BR and HR.

**Figure 9 sensors-22-04112-f009:**
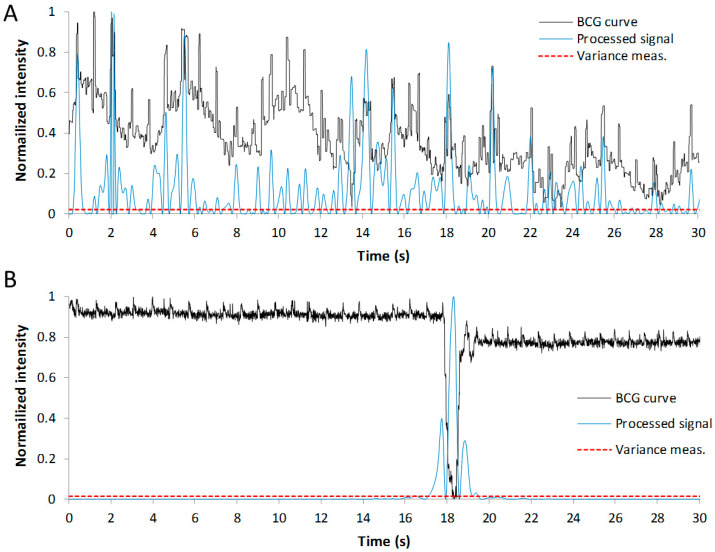
Clean (**A**) and corrupted (**B**) BCG curves (in black). Blue and red lines correspond to the data used for the prediction of movements in the sequence.

**Table 1 sensors-22-04112-t001:** Error for different bias currents.

Relative Error (%)				
I (mA)	R	G	B	IR
1	0.3	0.3	0.3	0.4
2	0.1	0.5	0.2	0.2
3	0.9	0.4	0.1	0.1
6	0.3	0.4	0.3	0.2
8	0.4	0.4	0.3	0.2
10	0.3	0.3	0.3	0.2

**Table 2 sensors-22-04112-t002:** Comparison of analog and digital stages of this work with recently published BCG systems.

Ref.	Extracted Parameters	Sensor	Analog Stages	Digital Processing
[19]	Movement, HR	Load cells	Instrumentation amplifier + low-pass filter + amplifier	Segmentation + variance measure + J-Peaks Detection + Wavelet Transform+ S/N ratio
[24]	HR, BR	Photodiode	High-pass filter + amplifier + low-pass filter + notch filter	Local interval estimation algorithm
[26]	BCG curves	EMFi	Amplifier + low-pass filter	Filtering
[28]	HR	EMFi	Amplifier + low-pass filter	Peak Detection + Filtering
[29]	HR, BR, apnea, snoring, movement	Pressure sensor	Amplifier + filters + envelope detector + control	FFT + standard deviation + S/N ratio
[30]	Movement	EMFi	Amplifier + inverter amplifier + low-pass filter	Filtering + variance measure + Neyman–Pearson detection rule + sequential detection rule
This work	HR, BR, movement	Digital photodetector	None	FFT + filtering + density of significative samples

## Data Availability

Not applicable.

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
