# Peer review of "Digital Optical Ballistocardiographic System for Activity, Heart Rate, and Breath Rate Determination during Sleep"

_sensors, 2022, doi:10.3390/s22114112_

Round 1
Reviewer 1 Report
In overall, it is interesting topic and approach.
- How can you tell the experiment method, e.g. using a small PCB with 5x5cm in size is proper and relevant to the proposed solution? Why not 50x50cm to be more practical?
- It lacks of profound and in-depth theorical concept and ideas for teh methods and results analysis. Describe why this approach and method are useful and significant comparing to other methods?
- All the figures don't show how relevant they are to the methods and experiment. What are you really trying to show by the figures without telling real meangingful values?
- It can include more comprehensive experiment and analysis from the perspective of thorical approach instead of 'looks like a simple project'
Some minor feedbacks in below:
The abstract describes too much of the method which can be included in other sections. Try to focus on balanced briefing
- brief introduction with problems
- methods
- results
- achievements
line35: This table.... => what is "This table"? describe the rigid table in detail.
38: fix the grammar - "they are a useful tool "
48: "other medical equipment" => provide an example.
56: too many unnecessary references, e.g. "Today, it is still one of its main objectives [19, 20]", "Therefore, many BCG 64
systems have been presented oriented to determining sleep parameters and sleep stage of 65
patients [25-28]. " => try to add useful and essential references only.
82: "As explained in Section I, the objective of this work is the development of a BCG system 82 suitable to be embedded in a bed mattress and able to provide estimations of the HR, BR 83 and motion of a lying patient making use of non-complex electronics.:" => remove this clause as it is repeated in section 1. why do you repeat and describe here?
Reviewer 2 Report
Thank you for sharing beautiful data. Let me ask the authors some questions.
1. What is the weight of the BCG system?
2. Are there any disadvantages to using a smart phone in the BCG system? For example, radio wave interference.
3. Are the subjects of this study healthy? Or is it some kind of patient?
4. How did the apneic occur be evaluated in relation to the above? Did a healthy person hold his breath? Or was it evaluated during sleep in patients with Sleep Apnea Syndrome? Also, have the authors been able to confirm the decrease in SpO2?
5. For example, I think this BCG system is useful for diagnosing Sleep Apnea Syndrome, but how many hours can it be recorded?
6. Is it possible that the BCG system will fall out due to turning over during sleep?
7. Can the BCG system be adapted to children?
Thank you again.
Reviewer 3 Report
Authors propose new BCG system for monitoring person during sleep. Sensor is quite original but there is several unclear statements. Authors use white LED and detector which measures independent R,G,B and IR components. Additionally sensor is digital one with I2C interface. Therefore response is not frequent as it supposed to be. Authors did not report why they use digital color sensor instead of photodiode (which can be wide spectrum and much faster) There are also light to voltage detectors, that can be used in place of such sensor.
In my opinion paper should be revised with better explanation why such sensor is used.
Round 2
Reviewer 1 Report
It seems too much to change the experiment model to reflect to practical situations with bigger size of board. It's still doubtful using 5x5cm against a human body to check turning for instance. There's an evidence of iprovement for other feedback explained.